# Treatment of Medial Instability of the Carpometacarpal and Tarsometatarsal Joints Using the Isolock^®^ System in Two Dogs

**DOI:** 10.3390/ani14040577

**Published:** 2024-02-08

**Authors:** Stefania Pinna, Chiara Tassani, Matteo Di Benedetto

**Affiliations:** Department of Veterinary Medical Sciences, University of Bologna, Via Tolara di Sopra 50, 40064 Ozzano dell’Emilia, Bologna, Italy; chiara.tassani2@unibo.it (C.T.); matteo.dibenedetto3@unibo.it (M.D.B.)

**Keywords:** carpometacarpal joint, tarsometatarsal joint, valgus instability, dog

## Abstract

**Simple Summary:**

Carpometacarpal and tarsometatarsal instabilities are rare diseases in dogs as compared to subluxations of the other joints of the carpus and the tarsus. Arthrodesis, skeletal external fixator use and ligament reconstruction are the most common surgical procedures for this pathology; however, high complication rates have been reported. This case report describes a novel procedure for the treatment of valgus instability of the carpometacarpal and tarsometatarsal joints using ultra-high-molecular-weight polyethylene suture (Isolock Intrauma^®^). The clinical and radiological findings are reported and a detailed description of the surgical technique is provided. The long-term outcomes reported in this study suggest that using a medial joint reinforcement system can be an easy, safe and effective treatment for joint instability.

**Abstract:**

This case report describes a novel procedure using the Isolock Intrauma^®^ implant system for treating medial instability of the carpometacarpal and tarsometatarsal joints, as demonstrated in in two dogs. A 9-year-old spayed female Spanish greyhound presented with a non-weight-bearing right hindlimb following a trauma. The clinical and radiological findings confirmed medial tarsometatarsal instability consistent with valgus deviation of the tarsus and the opening of the joint line on the medial aspect from the first to the third tarsometatarsal joints. A 4-year-old female Drahthaar presented with a non-weight-bearing left forelimb, swelling of the carpus and valgus instability. Radiological examination revealed a widening of the spaces between the intermedioradial carpal bone, second carpal bone and metacarpal bone II, confirming the medial carpometacarpal instability. In both cases, the Isolock system, an implant including ultra-high-molecular-weight polyethylene suture (UHMWPE), was used to reinforce the medial joint structures. Minor short-term complications were observed, such as swelling of the tarsal surgical site and hyperextension of the carpus, but these resolved spontaneously. No lameness or major complications were reported five months postoperatively. Carpometacarpal and tarsometatarsal instabilities are rare diseases in dogs as compared to subluxations of the other joints of the carpus and tarsus. There are no previous reports regarding the use of a UHMPWE implant for the treatment of these rare joint injuries, though the present case report suggests the validity and efficacy of the Isolock Intrauma^®^ implant for restoring carpal and tarsal stability and preserving joint mobility.

## 1. Introduction

The tarsus and the carpus of dogs are complex joints, the stability of which relies on ligaments, tendons and a retinaculum [1,2]. The antebrachiocarpal, middle carpal and carpometacarpal joints, as well as the tarsocrural, intertarsal and tarsometatarsal joints, form the joints of the carpus and tarsus, respectively [3]. Thus, numerous joint injuries can occur; antebrachiocarpal and tarsocrural dislocations/subluxations are the most common injuries due to the greater mobility of these joints, their distal location and limited tissue coverage [4,5,6,7,8]. On the other hand, carpometacarpal and tarsometatarsal subluxations are reported to be rare [1,2]. The medial collateral and the multiple interosseous ligaments, which provide support between the numerous carpal and tarsal bones, are easily susceptible to injuries compared to their lateral counterparts due to the physiological valgus of the distal joints [8,9].

Dogs usually show different degrees of lameness, swelling, pain and mediolateral instability of the affected joint [1,2,7]. Clinical examination and stress radiographs are required to confirm the diagnosis of injury to the medial compartment ligaments of a joint [5,8,10]. Pancarpal and pantarsal arthrodesis [8,11,12,13,14,15,16,17,18,19,20,21] and ligament reconstruction with synthetic implants [8,22,23,24,25,26] are two surgical treatments commonly suggested for these injuries. Transarticular immobilisation with an external skeletal fixator (ESF) or a calcaneotibial screw has often been associated with tarsal plate osteosynthesis or synthetic implants [13,14,23,24,27,28,29]. In the case of carpal instability, conservative management has been reported [30], such as rigid external coaptation using splints or casts associated with carpal constructs [7,31]. A high complication rate has been reported after carpal and tarsal arthrodeses, even when they are combined with other treatments [12,14,20,32,33,34,35].

Only a few studies have described the various synthetic implants used for treating collateral ligament injuries, both ex vivo [25] and in vivo [8]. Ultra-high-molecular-weight polyethylene (UHMWPE) suture material has been used in human and veterinary surgery [36]. In small animal surgery, the use of UHMWPE suture has been described for the extra- and intracapsular treatment of cranial cruciate ligament rupture, the fixation of radial head luxation in a Monteggia injury and the fixation of coxofemoral luxation [37,38,39,40]. A recent cadaveric study has shown the valuable mechanical properties of UHMWPE as compared to screw fixation in temporary tarsocrural joint immobilisation [36], and in vivo, UHMWPE has successfully been used for reconstruction of the medial collateral tarsal ligament in a dog with tarsocrural instability [8].

To the best of the authors’ knowledge, no reports exist concerning the use of UHMWPE for the treatment of valgus instability of the carpometacarpal and tarsometatarsal joints. The use of the Isolock implant (Isolock, Intrauma^®^, Rivoli, Torino, Italy) in two dogs is herein described, and long-term follow-ups are reported.

## 2. Materials and Methods

### 2.1. Case Description

Case #1: A 9-year-old spayed female Spanish greyhound, weighing 27 kg, presented with a non-weight-bearing right hindlimb, which occurred after a fall while running. Clinical examination showed swelling of the tarsus and valgus instability. Orthogonal and manual-stress radiographic views revealed an opening on the medial aspect of the joint line from the first to the third tarsometatarsal joint, consistent with valgus instability, and a fracture of the proximal base of metatarsal bone V (Figure 1). When considering the regional anatomy [3], these findings were suggestive of disruption of the fibres of the long part of the medial collateral ligament attached to metatarsal bones I and II, and of the small vertical ligaments that join the row of tarsal bones to the base of the metatarsal bones. Tarsometatarsal subluxation with medial instability, and an avulsion fracture at the insertion of the fibularis brevis and abductor digiti V muscles at the base of metatarsal bone V were diagnosed.

Case #2: A 4-year-old female Drahthaar, weighing 24 kg, presented with a non-weight-bearing left forelimb, which occurred after a presumed trauma. Clinical examination revealed soft-tissue swelling of the carpus and valgus instability. Orthogonal and manual-stress radiographic views revealed a widening of the spaces between the intermedioradial carpal bone, second carpal bone and metacarpal bone II [3]. The radiographic images were suggestive of distraction of the fibrous joint capsule and dorsomedial ligamentous disruption. These openings on the medial aspect of the carpus were consistent with valgus instability of the middle carpal and carpometacarpal joints. An oblique fracture of the body of metacarpal bone V was also diagnosed (Figure 2).

### 2.2. Design of Implant

The Isolock Intrauma^®^ implant is an orthopaedic system consisting of a braided ultra-high-molecular-weight polyethylene (UHMWPE) suture connected to a titanium button and stainless-steel needle. The Isolock kit consists of the implant, a titanium interference screw, a reamer and two Kirschner wires (K-wires). The manufacturer has indicated that UHMWPE braided suture is a very tough material, with the highest impact strength of any thermoplastic presently made. UHMWPE is odourless, tasteless, nontoxic, highly resistant to corrosive chemicals, has extremely low moisture absorption and a very low coefficient of friction and is self-lubricating and highly resistant to abrasion.

### 2.3. Surgical Technique

The owners were informed regarding the anaesthesiologic and surgical procedures and then signed an informed consent form. Each dog was anaesthetised according to standard protocols, in lateral recumbency with the injured limb positioned uppermost. The surgical site was prepared aseptically and the limb was draped.

Case #1: The medial aspect of the tarsometatarsal joint was accessed and the separation of the tarsal bones from the base of the second metatarsal bone was evident. The Isolock Intrauma^®^ implant was used to reinforce the medial aspect of the tarsometatarsal joint. Two bone tunnels were prepared. The first bone tunnel (tarsal tunnel) was drilled in a medial-to-lateral direction with a K-wire through the distal row of the tarsal bones. A second bone tunnel (metatarsal tunnel) was drilled from the medial side of the base of metatarsal bone II, obliquely, in a medio-proximal to latero-distal direction through the other metatarsal bones. The correct positioning and the direction of the bone tunnels were verified by intraoperative fluoroscopy. Each tunnel was over-drilled by inserting the reamer into the K-wire and drilling the hole to the opposite side in which a small skin incision was made. Then, the UHMWPE suture was passed through the tarsal tunnel in a lateral-to-medial direction and then through the metatarsal tunnel in a medial-to-lateral direction by using the connected Isolock stainless-steel needle. As a result, the titanium button rested on the cortical bone of the lateral aperture of the tarsal tunnel. The end of the suture was manually stretched from the lateral aperture of the metatarsal tunnel to the valgus correction, then fixed with the interference screw at the medial aperture of the same tunnel (Figure 3). The remaining suture was cut. Conservative management was chosen to treat the avulsion fracture of the proximal base of metatarsal bone V.

Case #2: A medial approach to the carpometacarpal joint was carried out, and instability was confirmed by identifying the widening of the medial space between the carpal and the metacarpal bones. The Isolock Intrauma^®^ implant was used to stabilise the joint utilising the same technique as described above; however, the bone tunnel accesses were obviously different. The first bone tunnel (radial tunnel) was drilled from the medial side of the distal radial epiphysis in a medial-to-lateral direction, and the second bone tunnel (metacarpal tunnel) was drilled from the proximal part of the body of metacarpal bones II, III and IV obliquely, in a medio-proximal to latero-distal direction. The suture was fixed with the titanium button at the lateral aperture of the radial tunnel and the interference screw at the medial aperture of the metacarpal tunnel. Fixation of the metacarpal bone V fracture was carried out using a 2.7 mm lag screw and a 1.2 mm K-wire (Figure 4).

In both cases, after surgery, the correct position of the Isolock implant was assessed using orthogonal radiographic projections, and the stability of the joints was checked by means of clinical tests and manual-stress radiographic views (Figure 5 and Figure 6). A Robert Jones bandage was applied for 2 weeks after surgery and the owners were instructed to restrict physical activity of their dogs for 30 days.

## 3. Results

Clinical and radiographic examinations were carried out at 15 days, 30 days and 5 months postoperatively (p.o.).

Case #1: The dog showed moderate lameness at 15 days p.o. and showed progressive improvement until complete resolution at the 30-day follow-up. The tarsometatarsal joint was stable and this was confirmed by stress radiography. At the first clinical follow-up, swelling of the tarsus and a serous fluid leak from the surgical site were reported. A bacterial culture was negative and these complications resolved spontaneously after 3 days. At five months p.o., no lameness or complications were reported, range of motion was normal and joint stability was preserved. The radiograph showed a partial pull-out of the screw, despite the fact that the radiographic stress view did not reveal any increase in the tarsal joint space. The fracture of metatarsal bone V was healed with callus (Figure 7).

Case #2: The owner reported poor compliance due to the exuberance of the dog. At 15 days p.o., the dog showed slight lameness and moderate carpal hyperextension (224° versus 193° of the contralateral limb). At 30 days p.o., slight carpal instability was detected, which was confirmed by a slight increase in the joint space as observed in the stress radiographic view. At five months p.o., the lameness had completely resolved, and the hyperextension of the carpus had improved (200° of the left limb versus 193° of the right limb). A radiograph of the stressed carpus showed a nearly normal joint space as compared to its counterpart. The fracture of metacarpus bone V had healed properly (Figure 8). 

## 4. Discussion

To the best of the authors’ knowledge, this is the first case report describing the use of UHMPWE for the treatment of medial instability of the carpometacarpal and tarsometatarsal joints. In the present paper, it was assumed that the use of the Isolock Intrauma^®^ implant as a synthetic suture could restore joint stability without interfering with joint mobility.

Joint instability occurs when the soft tissues that support the joint are unable to ensure normal functioning of the joint. Different types of tissue can be affected by injuries, namely muscles, tendons, ligaments and the synovial membrane. In the literature, dislocations/subluxations of the antebrachiocarpal or tarsocrural joints have been described more frequently than those of the carpometacarpal or tarsometatarsal joints. Partial or complete arthrodesis is often carried out to stabilise the carpal and the tarsal joints without, however, restoring the ligament function [11,12,13,14,15,16,17,18,19,20,21]. Some authors have recently suggested that ligament reconstruction could be a good surgical choice [8].

External trauma is the most frequent cause of injury to the carpus and the tarsus [4,5,6,8]. When shearing injuries occur, the placement of synthetic sutures, plates or screws is not recommended as they represent a bacterial pabulum, and there would be a high risk of infection. Therefore, the use of an ESF is suggested to stabilise the joint and treat soft-tissue injury [8,22,23,24,25,26]. However, as previously mentioned, an ESF has a high complication rate when treating dislocations/subluxations of the carpus and tarsus. In the present case report, the dogs presented medial instability with no shearing injury. For this reason, they were good candidates for synthetic-joint reinforcement to treat their valgus instability.

Surgical site infection is, however, the most feared complication when using synthetic implants [22,35]. In case #1, the postoperative swelling and leakage from the surgical wound were presumably related to soft-tissue inflammation, as the negative bacteriological examination excluded a surgical-site infection. At the five-month follow-up, neither dog showed any sign of inflammation or infection.

Tunnel ovalisation in metatarsal bone II and partial migration of the screw 30 days postoperatively were observed in case #1, as had been described in a previous study regarding the use of UHMWPE suture [8,25]. The partial loss of contact at the interface between the bone tunnel and the screw could lead to a loss of biomechanical strength of the construct, resulting in a loss of joint stability. Fortunately, this did not occur in case #1 in which there was no sign of instability. In this dog, the primary surgical stabilisation may have been reinforced by a subsequent biological stabilisation, which was able to maintain stability despite the partial failure of the implant [41,42].

In case #2, hyperextension of the carpal joint was observed 15 days postoperatively. Hyperextension usually occurs due to multiligamentous injury as the palmar carpal ligaments and the palmar interosseous ligament, the flexor retinaculum and the palmar fibrocartilage prevent hyperextension of the antebrachiocarpal joint [6,7]. In case #2, the dog was referred to the hospital with severe lameness, swelling and pain, and the lack of weight bearing did not allow hyperextension to be detected. Loss of palmar support can be hidden, with common and obvious causes for its failure to show, such as valgus instability or fractures [9]. Five months postoperatively, the carpal hyperextension was not completely resolved although it was significantly improved.

The cases described showed fractures of metatarsal bone V or metacarpal bone V. Some authors have considered internal fixation to be the most appropriate treatment for patients with other concomitant injuries, such as collateral ligament rupture or in the case of metatarsal bone II or V fractures [43,44]. In case #1, the medial joint instability was not attributed to the avulsion fracture of metatarsal bone V, and it was managed conservatively; in case #2, surgical treatment of the metacarpal bone V fracture using a screw and a Kirschner wire was chosen.

The present study has several limitations. The outcomes of only two cases are not adequate to evaluate the use of the Isolock implant; therefore, a large cohort of dogs should be recruited to validate the effectiveness and success of this implant and its associated surgical procedure for the treatment of carpometacarpal and tarsometatarsal instability. Another limitation is the lack of detailed dimensional anatomical imaging obtained by magnetic resonance imaging, which could have identified the ligaments involved. Proper knowledge of the injured ligament component would have provided accurate indications for the planning of the implant placement sites. The diagnosis was reached on the basis of the clinical picture and radiographic images by considering the regional anatomy.

## 5. Conclusions

In the two cases presented, the Isolock Intrauma^®^ suture was effective for the treatment of carpometacarpal and tarsometatarsal instability. It proved to be an easy-to-apply, resistant, elastic implant due to the inherent characteristics of UHMWPE; it could also be used to treat other common injuries involving the carpal and tarsal joints. This case report demonstrated that the Isolock implant can successfully restore carpal and tarsal stability, while avoiding the loss of joint mobility; however, additional studies are required.

## Figures and Tables

**Figure 1 animals-14-00577-f001:**
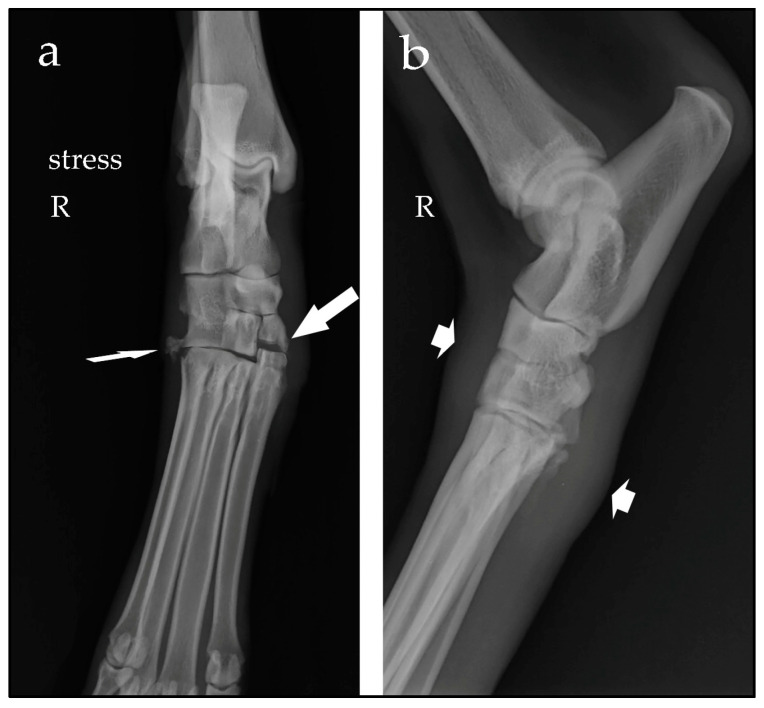
Stressed dorsoplantar radiographic view (**a**) and mediolateral view (**b**) of the right (R) tarsus. Note the opening of the tarsometatarsal joint line between the first to the third distal tarsal bones and the base of the metatarsal bones I to III (large arrow), the avulsion fracture of the proximal base of metatarsal bone V (narrow arrow) and the soft-tissue swelling (short arrows).

**Figure 2 animals-14-00577-f002:**
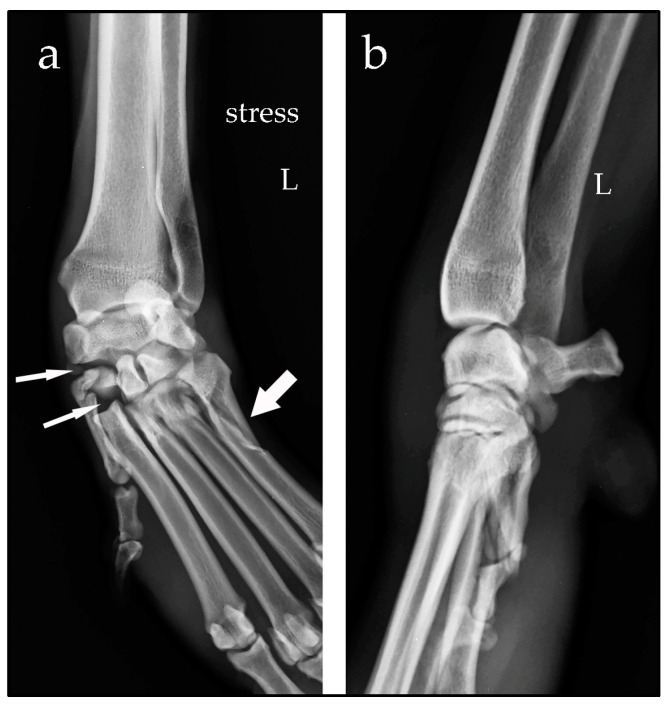
Stressed dorsopalmar radiographic view (**a**) and mediolateral view (**b**) of the left (L) carpus taken at the time of diagnosis. Note the opening of the spaces between the intermedioradial carpal bone, second carpal bone and metacarpal bone II (narrow arrows). An oblique fracture of the body of metacarpal bone V is evident (large arrow).

**Figure 3 animals-14-00577-f003:**
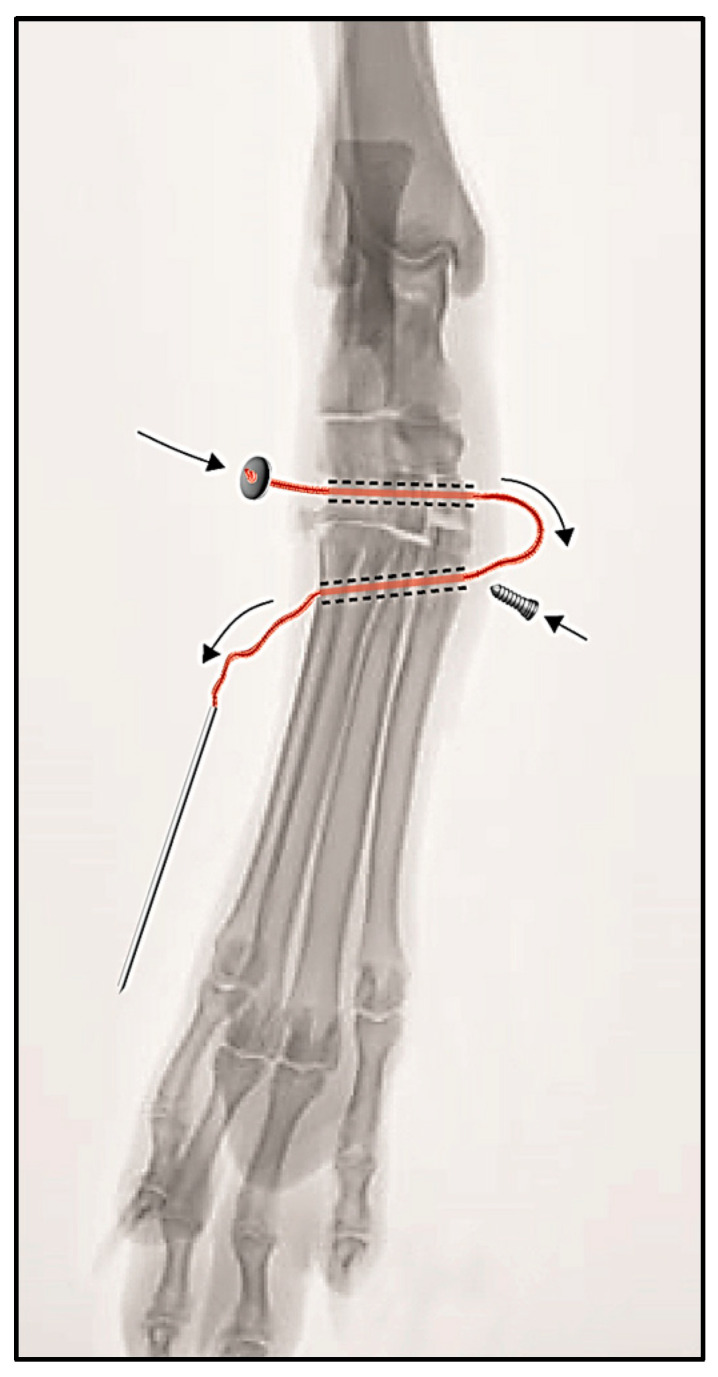
Schematic drawing of the surgical procedure for the treatment of tarsometatarsal instability. The black arrows indicate the direction of the insertion of the Isolock implant through the tarsal and metatarsal tunnels and the introduction of the interference screw.

**Figure 4 animals-14-00577-f004:**
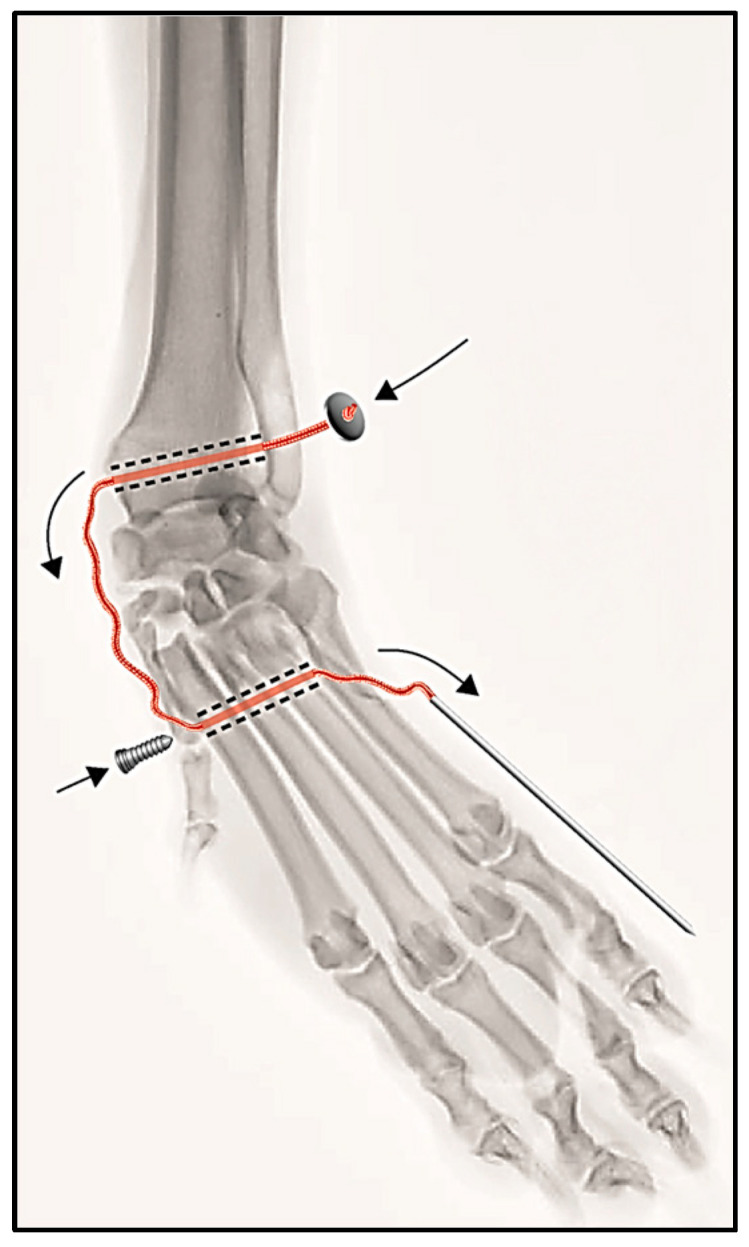
Schematic drawing of the surgical procedure for the treatment of carpometacarpal instability. The black arrows indicate the direction of the insertion of the Isolock implant through the carpal and metacarpal tunnels and the introduction of the interference screw.

**Figure 5 animals-14-00577-f005:**
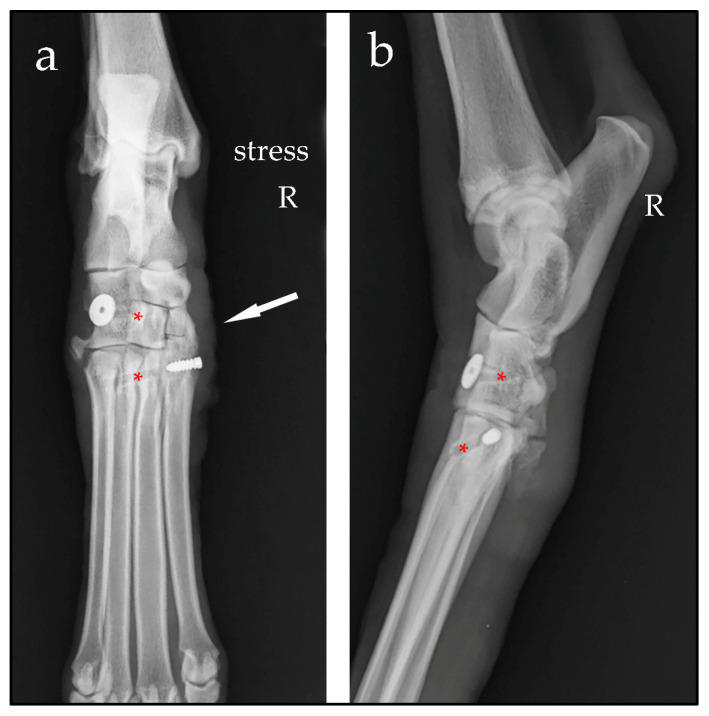
Stressed dorsoplantar radiographic view (**a**) and mediolateral view (**b**) of the right (R) tarsus taken postoperatively. The tarsometatarsal joint space has been restored to normal (arrow). The tarsal and metatarsal tunnels are visible (red asterisks); note the titanium button and interference screw placed at the end of the UHMWPE tensioned suture (not visible radiologically).

**Figure 6 animals-14-00577-f006:**
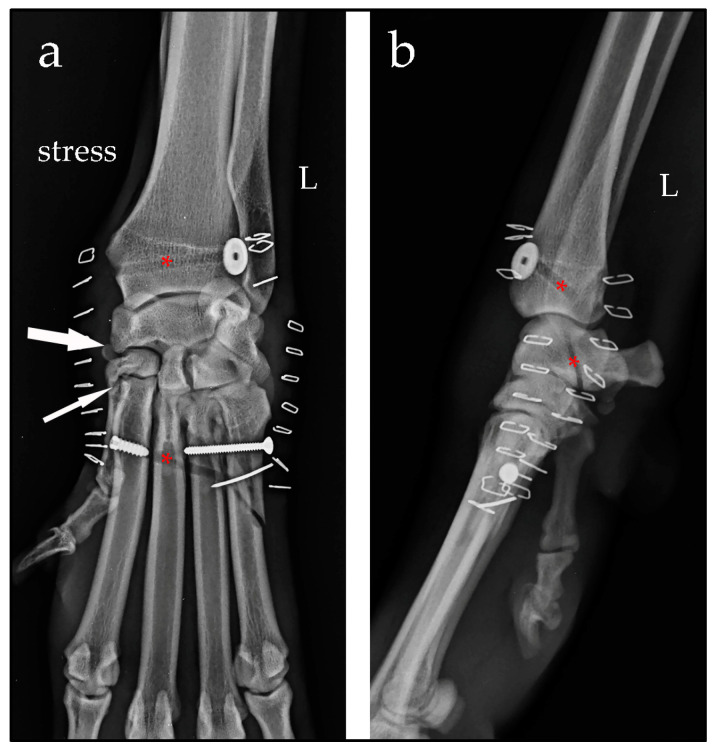
Stressed dorsopalmar radiographic view (**a**) and mediolateral view (**b**) of the left (L) carpus taken immediately after surgery. The joint space between the intermedioradial carpal bone and the second carpal bone is still slightly increased (large arrow); the joint line between the second carpal bone and metacarpal bone II is normal (narrow arrow). The carpal and metacarpal tunnels are visible (red asterisks); the titanium button and interference screw are in the correct position. The lag screw and the K-wire are correctly positioned to reduce the fracture of metacarpal bone V.

**Figure 7 animals-14-00577-f007:**
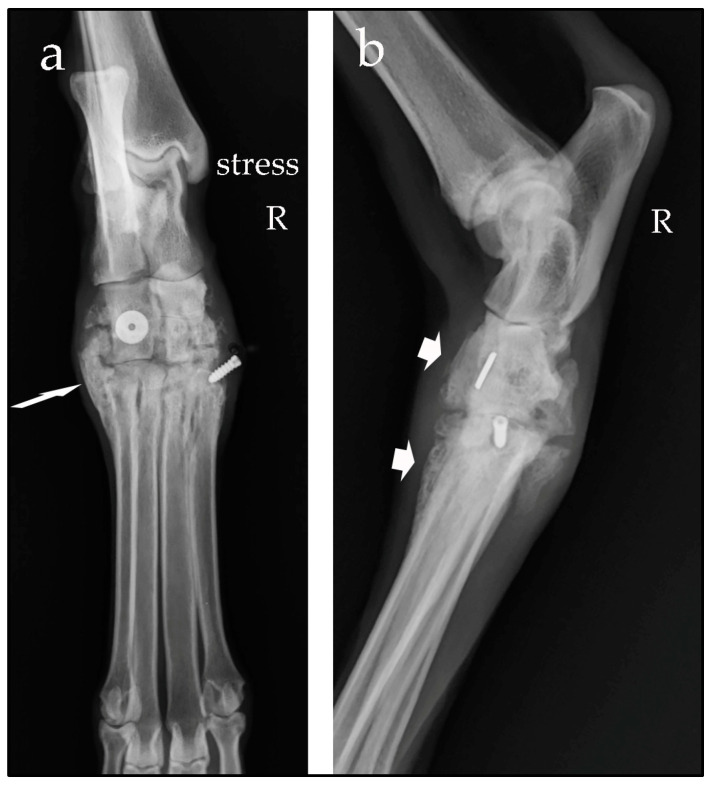
Stressed dorsoplantar radiographic view (**a**) and mediolateral view (**b**) of the right (R) tarsus taken five months after surgery. Radiographic images show a partial pull-out of the screw, slight soft-tissue swelling, and mild amorphous periosteal reaction of the dorsomedial aspect of the tarsometatarsal joint (short arrows). A bony callus is visible and is consistent with the healing of the avulsion fracture at the base of metatarsal bone V (narrow arrow).

**Figure 8 animals-14-00577-f008:**
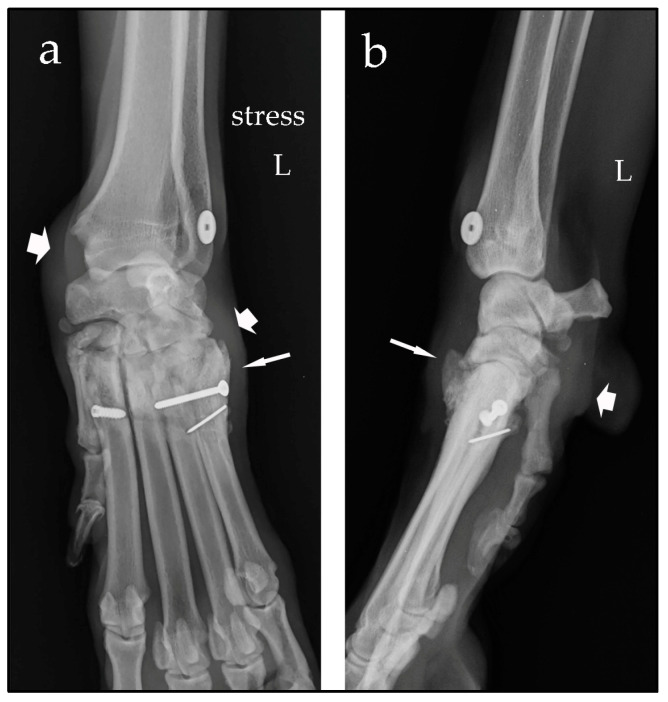
Stressed dorsopalmar radiographic view (**a**) and mediolateral view (**b**) of the left (L) carpus taken five months after surgery. Radiographic images show a nearly normal joint space, no sign of line fracture and a bony callus (narrow arrows). Mild soft-tissue swelling is noted (short arrows). No migration of the orthopaedic implants is observed.

## Data Availability

Data are contained within the article.

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
