# Peer review of "Treatment of Medial Instability of the Carpometacarpal and Tarsometatarsal Joints Using the Isolock® System in Two Dogs"

_animals, 2024, doi:10.3390/ani14040577_

Round 1
Reviewer 1 Report
Comments and Suggestions for Authors
Dear Authors,
the paper titled "Reinforcement of the carpometacarpal and tarsometatarsal joints of two dogs using the Isolock® system for the treatment of medial instability: a preliminary study" aimed to describe a novel procedure for treating medial instability of the carpometacarpal and tarsometatarsal joints in two dogs.
The paper looks interesting but needs a larger study population to be properly addressed.In addition, a longer follow up period, at least 12 months, should be considered.
For these reasons I think the paper cannot be accepted in the present form.
I hope you may increase the case numbers in order to better understand the inclusion criteria and the potential complications related to the procedure.
Kind regards
Author Response
Response: Dear Reviewer, thank you for the good evaluation of the paper. Regarding the small sample, this limitation was highlighted in the discussion section: see lines 262-266 " The outcomes of only two cases are not adequate to evaluate the use of the Isolock implant; therefore, a large cohort of dogs should be recruited to validate the effectiveness and success of this implant and its associated surgical procedure for the treatment of carpometacarpal and tarsometatarsal instability." … “additional studies are required”. It can be assumed that the aim of the study was to present a novel technique the good results of which, obviously partial, could suggest a new way of furthering large-scale applications.
According to the Animal’s guidelines, case reports must provide an in-depth, rather than superficial, review of a particular case. The purpose of this presentation should be that it suggests a novel way of interpreting existing knowledge on the topic. …. for this reason, 2 cases were described in detail in which, although two different joints were treated, the use of a new device for the same type of medial instability was presented. In addition, 5-month follow-ups were reported, which is enough time to describe the healing in joint instabilities, or 6 months as suggested by some guidelines. The Animals guidelines do not indicate the number of cases to be described.
In the literature, the majority of case reports describe 1 or 2 cases, often only the novel technique. permit me to list the literature reference: A case report is a detailed account of the symptoms, signs, diagnosis, treatment and follow-up of a single patient..... from: Guidelines To Writing A Clinical Case Report. Heart Views. 2017 Jul-Sep;18(3):104-105. doi: 10.4103/1995-705X.217857. PMID: 29184619; PMCID: PMC5686928.
Reviewer 2 Report
Comments and Suggestions for Authors
The case report presents a novel procedure for treating medial instability of the carpometacarpal and tarsometatarsal joints in the dogs. In general the manuscript is well written. The main limitation is too small number of animals. Thus the further study are necessary for the confirmation of obtained results.
Author Response

(The authors gave the same response as above.)

Reviewer 3 Report
Comments and Suggestions for Authors
The design of the work is simple and clear, the objective of stabilizing tarsometatarsal and carpometacarpal dislocations using an isolock device is timely.
The number of cases provided is very limited (1 for each region) but the description of the cases is very good, as are the results obtained.
The discussion allows us to compare this technique with others previously described, this technique being simple to apply and with good results.
Author Response

(The authors gave the same response as above.)
